


# Volcanic stratospheric sulphur injections and aerosol optical depth from 500 BCE to 1900 CE

Matthew Toohey[1], Michael Sigl[2,3]

[1]GEOMAR Helmholtz Centre for Ocean Research Kiel, Germany
[2]Laboratory of Environmental Chemistry, Paul Scherrer Institute, 5232 Villigen, Switzerland
[3]Oeschger Centre for Climate Change Research, 3012 Bern, Switzerland

*Correspondence to*: Matthew Toohey (mtoohey@geomar.de)

**Abstract.** The injection of sulphur into the stratosphere by explosive volcanic eruptions is the cause of significant climate variability. Based on sulphate records from a suite of ice cores from Greenland and Antarctica, the eVolv2k database includes estimates of the magnitudes and approximate source latitudes of major volcanic stratospheric sulphur injection (VSSI) events from 500 BCE to 1900 CE, constituting an update of prior reconstructions and an extension of the record by 1000 years. The VSSI estimates incorporate improvements to the ice core records in terms of synchronization and dating, refinements to the methods used to estimate VSSI from ice core records, and includes first estimates of the random uncertainties in VSSI values. VSSI estimates for many of the largest eruptions, including Samalas (1257), Tambora (1815) and Laki (1783) are within 10% of prior estimates. A number of strong events are included in eVolv2k which are largely underestimated or not included in earlier VSSI reconstructions, including events in 540, 574, 682 and 1108 CE. The long-term annual mean VSSI from major volcanic eruptions is estimated to be ~0.5 Tg [S] yr$^{-1}$, ~50% greater than a prior reconstruction, due to the identification of more events and an increase in the magnitude of many intermediate events. A long-term, latitudinally and monthly resolved stratospheric aerosol optical depth (SAOD) time series is reconstructed from the eVolv2k VSSI estimates, and the resulting global mean SAOD is found to be similar (within 33%) to a prior reconstruction for most of the largest eruptions. The long-term (500 BCE—1900 CE) average global mean SAOD estimated from the eVolv2k VSSI estimates and including a constant "background" injection of stratospheric sulphur is ~0.014, 30% greater than a prior reconstruction. These new long-term reconstructions of past VSSI and SAOD variability give context to recent volcanic forcing, suggesting that the 20$^{th}$ century was a period of somewhat weaker than average volcanic forcing, with current best estimates of 20$^{th}$ century mean VSSI and SAOD values being 25 and 14% less, respectively, than the mean of the 500 BCE to 1900 CE period. The reconstructed VSSI and SAOD data are available at doi:10.1594/WDCC/eVolv2k_v2.



## 1 Introduction

The injection of sulphur into the stratosphere by explosive volcanic eruptions has important ramifications for the Earth's climate. Sulphur-containing gases emitted by volcanic eruptions, including $SO_2$ and $H_2S$, lead to the formation of liquid sulphate aerosols. In the stratosphere, sulphate aerosols have a lifetime on the order of years. These aerosols scatter incoming
solar radiation and absorb infrared radiation, leading to a net decrease in radiation reaching the Earth's surface, and associated cooling (Robock, 2000).

Reconstructions of the history of climatic forcing by past eruptions have a long history (Lamb, 1970), and are an important ingredient in an understanding of past climate variability. Volcanic reconstructions have been extensively used to understand climate variability in instrumental and proxy-based climate records (Crowley et al., 2008; Hegerl et al., 2007; Masson-
Delmotte et al., 2013; Sigl et al., 2015), and have been used to show that volcanism is the dominant natural driver of climate variability in the Earth's recent past (Schurer et al., 2013). Volcanic reconstructions are also increasingly being used to understand the role of sudden climate changes in the evolution of past societies as recorded in documentary and archaeological archives (Ludlow et al., 2013; Oppenheimer, 2011; Toohey et al., 2016a).

Volcanic forcing reconstructions provide essential boundary conditions for climate model simulations which aim to
reproduce past climate variability. In one commonly used methodology, climate models take as input reconstructed volcanic forcing data sets, which prescribe certain physical aspects of the volcanic stratospheric sulphate aerosol. More recently, climate models have been coupled with prognostic aerosol microphysical modules, which allow the explicit simulation of the growth, transport and removal of stratospheric aerosols (e.g., English et al., 2013; Mills et al., 2016; Timmreck, 2012; Toohey et al., 2011). For these models, simulating the effects of volcanic eruptions on climate requires estimates of the
amount of sulphur injected into the stratosphere, and the time and location of that injection. Prognostic aerosol modeling comes however with an associated computational expense, and many model simulations continue to use prescribed aerosol forcing sets as input. The Easy Volcanic Aerosol (EVA) forcing generator is a simple and flexible module which produces stratospheric aerosol properties for use in climate models when given the sulphur injection magnitude, time and approximate location (Toohey et al., 2016b). EVA therefore provides the bridge necessary to allow different types of models to use time
series of volcanic stratospheric sulphur injection (VSSI) as a common basis volcanic forcing data set.

For eruptions since approximately 1979, VSSIs can be estimated based on satellite observations of the initial $SO_2$ plume (Bluth et al., 1997; Carn et al., 2016; Clerbaux et al., 2008; Guo et al., 2004; Höpfner et al., 2015; Read et al., 1993). Prior to the satellite era, information on the sulphur injection can be inferred from different records, including optical measurements (Sato et al., 1993; Stothers, 1996, 2001), geologic information on the on the eruptive magnitude and volatile content of the
erupted magma (Metzner et al., 2012; Scaillet et al., 2003; Self, 2004; Self and King, 1996), and ice cores (Ammann et al., 2003; Clausen and Hammer, 1988; Robock and Free, 1995; Zielinski, 1995). Ice cores in particular provide long records with unequalled temporal accuracy and precision of volcanic eruptions from around the globe (Cole-Dai, 2010; Robock and Free, 1995).





Ice cores were first used to estimate the stratospheric sulphate aerosol mass burden following explosive volcanic eruptions by Clausen and Hammer (1988). Zielinski (1995) used similar techniques to estimate the sulphate aerosol loading resulting from eruptions spanning 2100 years, based on chemical analysis of the Greenland GISP2 ice core. Robock and Free (1995) constructed an index of volcanic climate forcing from a compilation of multiple ice cores from numerous NH and SH polar

ice cores. Based on a larger compilation of polar ice cores, Gao et al. (2008) reconstructed both stratospheric sulphur injections and reconstructions of the spatiotemporal spread of aerosol mass in the stratosphere over the time period 501-2000. Other volcanic reconstructions (e.g., Ammann et al., 2003; Crowley and Unterman, 2013) have provided estimates of the radiative impacts of past eruptions based on analysis of ice cores, without explicitly estimating sulphate aerosol masses or sulphur injections.

Recent improvements to the ice core record of past volcanism (Plummer et al., 2012; Sigl et al., 2014, 2015) motivate a revision and extension of sulphur injection reconstructions. The record of volcanism preserved in Antarctic ice has been improved based on the compilation of a larger set of ice cores, extending back to 500 BCE, with a largely static number of cores used to compile an Antarctic average for the past 2,000 years (Sigl et al., 2014). An adjustment to the age model used to date past volcanic events has resulted in better agreement between ice core sulphate signals and cooling signals preserved

in tree rings, improving confidence in the accuracy of the new ice core record dating (Sigl et al., 2015).

This manuscript describes the construction of a new sulphur injection database from ice core records, based on newly compiled ice core sulphate composites. It presents the justification for a modification to the method used to convert ice core sulphate fluxes to VSSI compared to past works. Finally, we present estimates of SAOD, translated from the VSSI estimates presented here using the EVA forcing generator, and compare the resulting record with prior reconstructions. Together, the

resulting VSSI and SAOD datasets represent significant updates and improvements to the volcanic forcing datasets (Crowley and Unterman, 2013; Gao et al., 2008) used in numerous prior climate model simulations, including those used in the Paleo-modelling Intercomparison Project (PMIP) Phases 2 and 3 (Schmidt et al., 2011).

## 2. Method

### 2.1 Ice core data

Sulphate (or sulphur) in ice cores can have several sources, of which marine biogenic emissions of dimethyl sulphide and volcanic sulphur emissions are the most important contributions during the preindustrial era (Cole-Dai, 2010). All reconstructions of volcanic sulphate mass deposition from ice-cores agree in the general methodology, in which the non-volcanic (or background) contribution to the total sulphate (or sulphur) at the ice-core site is assumed to be slowly varying, and thus can be well approximated by simple functions such as splines, running medians, or a constant value. Upon

subtracting the estimated background from the total concentrations, the remaining excess concentrations are attributed to volcanic origin (Gao et al., 2008; Gao et al., 2006; Sigl et al., 2014; Traufetter et al., 2004; Zielinski, 1995).



### 2.1.1 Ice core sulphate composites

As the basis of the volcanic reconstruction, we used an existing compilation of synchronized volcanic sulphate records from ice-cores in Greenland and Antarctica (Sigl et al., 2015), complemented with the GISP2 ice-core record from Greenland (Zielinski et al., 1996) to improve the sampling density especially during the earlier period of our reconstruction. The

compilations use only sulphur and sulphate ($SO_4^{2-}$) concentration measurements, excluding measurements of electrical conductivity or acidity since other species than sulphur (e.g. chlorine, fluorine, nitrate, carboxylic acids) may also contribute to the total measured acidity (Clausen et al., 1997; Pasteris et al., 2014). A list of ice cores used is included in Table S1. In summary, the reconstruction is based on ice cores from three sites in Greenland, including NEEM (Sigl et al., 2014, 2015), NGRIP (Plummer et al., 2012) and GISP2 (Zielinski, 1995; Zielinski et al., 1996); and from between 8 and 17 individual ice

cores from Antarctica included in the AVS-2k composite over the Common Era (Sigl et al., 2014), extended to earlier dates by the WDC (from 394 BCE) and B40 (from 500 BCE) ice cores (Sigl et al., 2015).

Volcanic sulphate flux is non-uniform over Greenland and Antarctica, and the shortest ice-core records (in time) usually have the largest deposition rates (Gao et al., 2007; Sigl et al., 2014). Taking account of changes in the sample size and pattern over time is a challenging issue in the construction of long-term ice sheet average fluxes. To simplify this process,

and improve the long-term stability of the resulting records, we preferentially used the longest ice core records currently available, resulting in a relatively constant sample size over most of the Common Era (N≥12; 200-1900 CE). With the addition of GISP2, only one ice core record has been included into the composite which was not part of the Sigl et al. (2015) database, but which has been used in previous compilations (Crowley and Unterman, 2013; Gao et al., 2008). Synchronization to the NS1-2011 chronology was performed against the NEEM sulphur record and 85 common stratigraphic

age markers (averaging one every 30 years) have been identified (Fig. S1-3). Over the past 2,500 years, the GISP2 sulphate record contains some missing data sections encompassing in total 160 years (6%), including the time period 532-606 AD which includes some of the largest volcanic eruptions in historic times.

Our estimates of the timing of volcanic sulphate enrichment in the ice cores are based on the most recently updated ice-core chronologies for Greenland and Antarctica (Sigl et al., 2015, 2016) and represent the first year in which a sulphate anomaly

was detected in the glacio-chemical records. Small adjustments were made in cases in which bipolar eruptions had slightly different ages in Greenland and Antarctica to derive a unified chronology (Sigl et al., 2015). Since the Greenland ice-core chronology (NS1-2011) was constrained with more absolute age markers than that from Antarctica (WD2014) a stronger weight was usually given to the Greenland ages. For known historic eruptions—some of which are verified by identifying and characterizing tephra in the ice core (Abbott and Davies, 2012; Jensen et al., 2014; Sun et al., 2014) - the exact timing

(calendar date, month, season) of the eruptions were used. For the majority of the volcanic deposition events over the past 2,500 years, the exact timing of the eruption cannot be constrained with the ice-core records alone, since volcanic sulphate has different atmospheric residence times depending on such details as the latitude, season and injection height of the





eruption. Thus the time lag between stratospheric injection at the source and deposition at the ice-core site can vary between a few weeks up to a year (Robock, 2000; Schmidt and Robock, 2015; Toohey et al., 2013).

For each ice core used, $SO_4$ mass deposition (or "flux", in units kg km$^{-2}$) was previously estimated for all volcanic events exceeding a predefined detection threshold by integrating the sulphate flux exceeding the natural background over the time

span of its deposition (Plummer et al., 2012; Sigl et al., 2013; Sigl et al., 2014; Sigl et al., 2015; Zielinski, 1995; Zielinski et al., 1996). While major volcanic signals related to eruptions such as Samalas (1257), Tambora (1815), Huyanaputina (1600) or Krakatao (1883) are clearly identified in all ice cores taken from a larger network, the smaller volcanic events may in some cases not be picked up by all ice-cores equally. For example, for the GISP2 ice core which was measured at biannual-resolution, Zielinski et al., (1995) detected fewer volcanic events than in comparably long, but higher resolved ice-cores

from NEEM and NGRIP. Low annual snowfall rates present in some areas in Antarctica can also occasionally lead to post-depositional changes of the original volcanic sulphate signature, as was shown in the example of the Tambora 1815 event using five ice cores all from Dome C (Gautier et al., 2016). But even under such extreme climate conditions present at some areas of East Antarctica, the loss of volcanic signatures from individual ice-core records by wind erosion is rather the exception than the rule, given that the major volcanic signatures of even much smaller events are continuously captured in

virtually all ice-cores over Antarctica (Sigl et al., 2014). The large number of ice-cores included in the AVS-2k network not only allowed firm detection of false positives and more precise quantification of total sulphate flux, but also a reduction of the detection limit and the identification of additional events that would not exceed the threshold limits set for a single sulphate record. By using an alternative detection approach (AVS-2k$_s$) based on extracting the event flux directly from a stacked sulphate concentration record characterized by increased signal-to-noise ratios compared to the more noisy

individual ice-core records, Sigl et al., (2014) extracted an additional 46 volcanic events with sulphate fluxes as low as 1-4 kg km$^{-2}$, whereas the detection limit for the individual ice-core sulphate series were close to 3-5 kg km$^{-2}$. With no comparable large network available from Greenland, the detection limit for the three Greenland ice cores in preindustrial times is typically in the range of 3-6 kg km$^{-2}$. After approximately 1900 AD, increased anthropogenic release of $SO_2$ into the troposphere from industrial processes masks many of the volcanic signatures during the 20[th] century in Greenland (Fig. S3).

For this reason, we have constrained the present reconstruction to the period before 1900: for the period thereafter, we recommend the use of estimates utilizing larger networks of Greenland ice cores (Crowley and Unterman, 2013; Gao et al., 2008) or from other multi-proxy reconstructions (Neely and Schmidt, 2016).

Antarctic and Greenland ice sheets average flux values were then computed based on the available ice core measurements for each volcanic event. In the AVS-2k compilation, individual ice cores were weighted in the overall average in order to

account for the spatial variability of deposition over the ice sheet. This was accomplished by first averaging flux values for pre-defined regions or depositional regimes, and then averaging the values over the different regions (Sigl et al., 2014). For Greenland, a simple average of the three ice cores was used, since the ice cores show no evidence of systematic differences in their measured values for 48 volcanic events common to all three cores (Fig. S4). In cases when volcanic sulphate was not detected in all three Greenland cores, sulphate flux was set to 50% of the detection limit for those ice cores without a strong



signal. This is motivated because visual inspection of the data for such cases often revealed the presence of a volcanic signal which did not exceed the detection threshold. These cases only include comparably small volcanic signals: the largest 40 events recorded in the Greenland ice cores since 200 CE all have signals in all three ice cores (providing no gaps in the records). Apparently, the detection threshold had been set differently for the individual ice cores so that many volcanic

events detected in NEEM and/or NGRIP have not been extracted for the GISP2 record. NEEM also seems to extract slightly fewer events than NGRIP, potentially owing to the greater background variability of sulphur, due to the closer proximity of the ice-core site to the ocean emitting biogenic sulphur species. Situated in the centre of Greenland, the NGRIP record has arguably the best ability to capture the atmospheric excess sulphate content from volcanic eruptions and we thus argue that the simple arithmetic mean is the most realistic metric to describe the true sulphate deposition over Greenland. For

Antarctica, occasionally missing values in individual cores have also not been set to "zero" prior to averaging, but the influence on the overall composite value is minimal, as is evident by the comparison with the alternatively stacked composite AVS-2k$_s$ (Sigl et al., 2014).

**2.1.2 Ice core uncertainties**

Uncertainty in the timing of volcanic events in ice core records arises from uncertainties in the annual-layer interpretation

when establishing the layer counted chronologies. Absolute age uncertainties in the ice core records used in eVolv2k are believed to be on average better than ±2 years during the past 1,500 years, and better than ±5 years some 2,500 years ago based on comparison to well dated tree-ring records (Adolphi and Muscheler, 2016; Sigl et al., 2016; Sigl et al., 2015).

Since the composites for Greenland (in general) and Antarctica (prior to the Common Era) are based on a few ice-cores only, we explore in the following how well large-scale sulphate deposition over Greenland and Antarctica are reproduced in

individual records or pairs of ice-core records. For this we use 48 events that are common to all three ice-core records in Greenland, and 48 events that are at least recorded in 10 ice-core records from Antarctica. In both cases, this threshold is more or less equivalent to eruptions with more than 6-7 kg km$^{-2}$ average sulphate deposition (Fig. S4).

For Antarctica, individual ice-core records such as B40 ($R^2$=0.88) and WDC ($R^2$=0.91) are strongly correlated with the ice-sheet wide deposition values based on a large number (N≥10) of individual ice cores (Fig. S4). A composite of only the B40

& WDC records—which is the sample used here prior to the Common Era in Antarctica–produces quite reasonable agreement with the full Antarctic composite for the 48 common events, with a correlation of $R^2$=0.93. Similarly, for Greenland, a composite of NEEM & GISP2 shows close correlation ($R^2$=0.98) with the 3-ice core Greenland composite, and correlations of single ice-core values versus the full composite are only slightly smaller, with GISP2 ($R^2$=0.76) showing the weakest agreement with the overall composite and NGRIP ($R^2$=0.89) having the highest correlation. The high level of

correlation - especially over Antarctica, where a large number of individual records is contributing to the composite - indicates that individual ice-core records are able to capture a large portion of the large-scale sulphate deposition.

Uncertainties in the Antarctic composite sulphate fluxes in AVS-2k are taken as reported by Sigl et al. (2015). From 1-2000 CE, the uncertainty in the Antarctic mean is quantified by the standard error of the mean (SEM) of the individual ice core





flux values. Typical (root-mean-square) uncertainties for this period in the Antarctic composites are approximately 13%. Before 1 CE, when only two ice cores are used in the construction of the Antarctic composite, a constant uncertainty value of 26% is assumed, based on regression analysis between AVS-2k (the ice-sheet average) and single ice-core records (e.g. WDC or B40, see Sigl et al., 2015).

Special consideration is paid to uncertainties in the Greenland composite sulphate flux due to the small sample size. Assuming that the fluxes retrieved from each ice core represent the true ice sheet average plus some normally distributed independent random error, estimates of the error variance for each ice core can be estimated (Appendix A). This analysis produces estimates of 46%, 45% and 33% for NEEM, NGRIP and GISP2 respectively (Fig. S5). Using standard error propagation rules, we estimate that when all three ice cores are used in an ice sheet composite, the resulting uncertainty is

approximately 22%, and two-core composites take uncertainties of approximately 32% (NEEM plus NGRIP), 28% (NEEM plus GISP2) and 28% (NGRIP plus GISP2).

## 2.2 Injection locations and dates

The locations and dates of stratospheric sulphur injections can be assigned based on matching ice core sulphate signals with observed historic eruptions. Here, following prior works (e.g., Crowley and Unterman, 2013; Plummer et al., 2012; Sigl et

al., 2013) we use the Volcanoes of the World online eruption database (Global Volcanism Program, 2013) and other sources of information to assign locations and dates to a number of events in the combined Greenland and Antarctic sulphate event inventory (Table 1). Such matches carry some degree of uncertainty and subjectivity. In some cases, the matches can be made with a high degree of confidence, based on temporal coincidence of exceptional sulphate signals with similarly exceptional eruptions, e.g. Tambora (1815) and Laki (1783). Chemical analysis of tephra extracted from ice cores has been

used to strengthen the matches for cases like Samalas (1257, Lavigne et al., 2013) and Changbaishan (946, Sun et al., 2014). In other cases, matches are based on little more than temporal coincidence between an ice core sulphate signal and an identified major eruption – such matches are prone to reexamination when ice core timescales are adjusted or the eruption catalogue is updated. For this exercise, we have attempted to err on the side of caution, and have discarded some matches used in prior works. For example, the large mid-15[th] Century ice core sulphate signal originally attributed to 1452/53 (Gao et

al., 2006) and recently refined by independent annual-layer counting to 1458 (Plummer et al., 2012; Sigl et al., 2013) often attributed to the Kuwae caldera, Vanuatu (17°S) is considered unidentified here.

For the remaining majority of ice core sulphate signals which are not easily matched to a known eruption, approximate eruption latitudes are assigned based on the presence, or lack of simultaneous signals in both Greenland and Antarctic ice core composites. Signals occurring synchronously (given small possible dating uncertainties) in both Greenland and

Antarctic composites are attributed to tropical eruptions, while those with signals in only one hemisphere are assumed to be extratropical in origin following (Sigl et al., 2015). We adopt a convention of assigning unknown tropical eruptions a latitude of 0°N, and extratropical eruptions a latitude of 45°N or S. Consistent with Crowley and Unterman (2013) unknown eruptions are assigned an eruption date of January 1.



### 2.3 Stratospheric sulphate injection estimation

The mass of sulphur injected into the stratosphere by an eruption ($M_S$) is eventually deposited to the Earth's surface. Assuming all injected sulphur is converted to sulphate aerosols before deposition, the mass of total sulphate deposition to the

surface ($D_{SO_4}$) is simply $3M_S$ due to the ratio of the molecular mass of $SO_4$ to the atomic mass of sulphur. From ice cores, the sulphate "flux" ($f_{SO_4}$) is derived, which represents the local accumulated sulphate mass density, in units of kg km$^{-2}$. If this flux was uniform over the Earth, estimating $D_{SO_4}$ (and thereby $M_S$) simply require multiplying the flux by the surface area of the Earth. Since the deposition is not spatially uniform, a transfer function is required to convert flux values from any location or area to estimates of $D_{SO_4}$, to account for the spatial inhomogeneity of deposition (Gao et al., 2007; Toohey et al.,

2013). Assuming that the deposition pattern is consistent for all events, for any location or region on Earth, a transfer function $L^{\text{global}}$, can be defined as

$$L^{\text{global}} = \frac{D_{SO_4}}{f_{SO_4}} = \frac{3M_S}{f_{SO_4}}. \tag{1}$$

Clausen and Hammer (1988) derived the first global transfer functions based on analysis of the radioactive products of nuclear weapons testing (NWT) in the 1950's and 1960's, with $L^{\text{global}}$ defined by the ratio of estimated release of radioactive

material and estimates of the radioactive flux from analysis of ice cores. Clausen and Hammer (1988) then used the global transfer functions to estimate the sulphate aerosol loading from a number of past eruptions, applying the transfer function for each ice core individually to the volcanic fluxes for each core, and averaging the result for a best estimate.

Gao et al. (2007) suggested that the sulphate flux to Greenland and Antarctica can be used separately as proxies for the NH and SH sulphate loading. In this methodology, transfer functions are required to connect the ice sheet sulphate fluxes from

Greenland and Antarctica ($f^G$, $f^A$, respectively, with the subscript "$SO_4$" discarded hereafter for brevity) to the total hemispheric deposition ($D^{NH}$ and $D^{SH}$). Defining a variable $\alpha$, which represents the ratio of NH to global sulphate deposition, and therefore the proportion of the total sulphur injection $M_S$ which is transported to and deposited over the NH:

$$\alpha = \frac{D^{\text{NH}}}{D^{\text{global}}} = 1 - \frac{D^{\text{SH}}}{D^{\text{global}}} \tag{2}$$

we can write transfer functions for the ice sheets of each hemisphere:

$$L^{\text{G}} = \frac{D^{\text{NH}}}{f^{\text{G}}} = \frac{3\alpha M_S}{f^{\text{G}}} \tag{3}$$



$$L^A = \frac{D^{SH}}{f^A} = \frac{3(1-\alpha)M_S}{f^A} \tag{4}$$

From Eq.s (3) and (4), we can write an expression for $M_S$ as a function of the measured Greenland and Antarctic fluxes and the transfer functions:

$$M_S = \frac{L^G f^G}{3} + \frac{L^A f^A}{3} \tag{5}$$

While the hemispheric partitioning coefficient α is not required to calculate $M_S$ via Eq. (5), it can be used as a proxy for the hemispheric asymmetry of the volcanic radiative forcing. In practice, the eVolv2k database includes the ratio ($R$) of the estimated NH-to-SH sulphate deposition:

$$R = \frac{L^G f^G}{L^L f^L} \tag{6}$$

where $R$ is simply related to α as $R = \alpha/(\alpha - 1)$.

Gao et al. (2007) derived values of the hemispheric transfer functions $L^G$ and $L^A$ separately for tropical and extratropical eruptions. For tropical eruptions, (Gao et al., 2007) used measurements of nuclear radioactivity from ice cores (Clausen and Hammer, 1988) and revised estimates of the stratospheric injection of radioactive fallout from NWT. Since the partitioning of radioactive material between the NH and SH after the NWT in the tropics is uncertain, Gao et al. (2007) assumed that between 1/2 to 2/3 of the radioactive material was transported into the Northern Hemisphere, (i.e., $0.5 < \alpha < 0.66$), which lead to estimates for $L^G$ ranging from 0.75-1.0×10$^9$ km$^2$. For $L^A$, Gao et al. (2007) used estimates of the sulphur injection by the 1991 eruption of Pinatubo, and measured sulphate in Antarctic ice cores. We revisit that calculation here, based on updated data. According to analysis of satellite retrievals (Guo et al., 2004), Pinatubo injected $18 \pm 4$ Tg of $SO_2$ into the lower tropical stratosphere, amounting to $9 \pm 2$ Tg [S]. Satellite records also show a fairly even transport of aerosol between the NH and SH, suggesting $\alpha \approx 0.5$. The Antarctic average sulphate flux following Pinatubo is approximately 11 kg km$^{-2}$ (Crowley and Unterman, 2013; Sigl et al., 2014). Using these values in Eq. (4) leads to an estimate of $L^A$ of $1.2 \pm 0.3\times10^9$ km$^2$, or approximately 0.9-1.5×10$^9$ km$^2$. This calculation is only slightly changed when considering the potential impact of the August 1991 Cerro Hudson eruption in Chile, which injected an estimated 0.75 Tg [S] into the stratosphere (Bluth et al., 1997). In this case, again noting that the observed SAOD after Pinatubo was balanced between the NH and SH, we infer that the SH loading was about half of the total sulphur injection by Pinatubo and Cerro Hudson, around $4.9 \pm 1$ Tg [S], which leads to a value for $L^A$ of $1.3 \pm 0.3 \times10^9$ km$^2$.

The hemispheric transfer function estimates from the NWT results (for Greenland) and the Pinatubo case study (for Antarctica) are both consistent with a value of 1.0×10$^9$ km$^2$. There is no reason that the transfer functions should need to be



the same for both Greenland and Antarctica—in fact given the spatial variability of simulated sulphate deposition patterns over the globe (Gao et al., 2007; Toohey et al., 2013), it would be perhaps surprising that the values are the same. On the other hand, ice core derived flux estimates for identified tropical eruptions cluster around having equal values for Antarctica and Greenland (Toohey et al., 2016a); therefore applying equal weight to the two hemispheric ice sheets in the estimation of

the global total seems to be a justifiable simplification.

A transfer function value for tropical eruptions of $1.0 \times 10^9$ km$^2$ is numerically identical to that derived by Gao et al. (2007), however, there is a subtle difference in our implementation. Our interpretation is that this transfer function relates the sulphate flux values with atmospheric sulphate mass (either the hemispheric sulphate deposition $D_{SO_4}$, or equivalently the theoretical maximum sulphate mass loading $3M_S$). In contrast, Gao et al. (2007) used the same value to estimate the sulphate

aerosol mass loading, which is different since sulphate aerosols include not just the mass of sulphate, but also that of water, as sulphate aerosols in the stratosphere are usually assumed to be 25% water by mass. To calculate the mass of sulphate aerosols from the mass of sulphate requires scaling by a factor of 4/3, therefore, our revised transfer function estimate is effectively 33% larger than that of Gao et al. (2007).

For extratropical eruptions, Gao et al. (2007) introduced a transfer function of $L^G = L^A = 0.57 \times 10^9$ km$^2$, based on analysis of

ice core radioactivity resulting from NWT at high latitudes, and on output from volcanic sulphate transport simulations with a general circulation model. The use of a smaller transfer function for extratropical eruptions seems appropriate since a larger proportion of the sulphate from such eruptions is likely to be deposited in the extratropics compared to tropical eruptions, necessitating a smaller transfer function to estimate the global deposition (or injected mass). We retain the value of $0.57 \times 10^9$ km$^2$ here, but again, interpret it rather as a transfer function between ice core-derived sulphate flux and sulphate loading (not

sulphate aerosol loading), producing an effective 33% increase in the transfer function compared to that of Gao et al. (2007). The threshold latitude separating tropical and extratropical eruptions is set to ±25°, based on satellite-based estimates of the "edges" of the stratospheric tropical pipe (Neu et al., 2003).

While it is thought that the majority of sulphate aerosol arising from extratropical eruptions is contained within the hemisphere of the eruption (Oman et al., 2006), it does seem possible that in some cases, sulphate from extratropical

eruptions may cross the equator in large enough quantities to be recorded in the ice sheets of the opposite hemisphere. Modelling results suggest that extratropical eruption can lead to ice sheet flux in the opposite hemisphere of the eruption around 10% that of the hemisphere of eruption, which are likely to produce detectable signals only for the largest such eruptions (Toohey et al., 2016a). In such cases, we used the extratropical transfer function to estimate the sulphate injection of the hemisphere of the eruption, and the tropical transfer function to estimate the injection to the other hemisphere.

It should also be noted that the method introduced above assumes that the ice core sulphate flux is directly proportional to the stratospheric injection. In reality, some of the sulphate deposited to ice sheets may come from volcanic sulphur emissions into the troposphere. Of particular importance are effusive (i.e., non-explosive) eruptions from Iceland, which, under the right meteorological conditions, may produce large sulphate deposition over Greenland, even when the stratospheric injection is minimal. Crowley and Unterman (2013) adjusted the Greenland flux values for the 1783 Laki eruption, deriving



a ratio of stratospheric-to-total sulphate flux of 0.15 based on analysis of the "far-field" Mt. Logan ice core. The proportion of Laki's stratospheric sulphur injection is indeed highly uncertain (Lanciki et al., 2012; Schmidt et al., 2012), and to date, little quantitative information on the stratospheric-to-tropospheric partitioning of sulphur injection is available for other Icelandic eruptions of the past 2500 years. Geological records suggest that purely effusive eruptions in Iceland are rare, and

that the eruption of Laki was characterized by both explosive and effusive phases (Thordarson and Larsen, 2007). Until an objective criterion can be established to quantify the proportion of ice core sulphate representing the stratospheric sulphate burden, we have chosen to maintain the assumption that all sulphate is stratospheric, but stress that this is a rather important potential source of uncertainty.

### 2.3.1 Uncertainty in VSSI

The VSSI estimates carry significant uncertainty, due to errors in the ice core flux measurements, and, more importantly, uncertainties in the transfer functions used to convert the ice core flux composites to estimates of VSSI.

Systematic uncertainties describe potential errors that are static, leading to overall bias in the estimated quantity. The most likely source of systematic error in the VSSI estimates comes from the uncertainty in the transfer functions $L^G$ and $L^A$. For $L^G$, the uncertain distribution of NWT radioactive material between the NH and SH leads to an uncertainty of ~15%,

although this uncertainty could be larger if one allows for the possibility of a larger range of possible hemispheric partitioning ratios. Uncertainty in $L^G$ is also strongly connected to uncertainties in the amount of radioactive material released by NWT, and its partitioning between the stratosphere and troposphere. For $L^A$, uncertainty arises due to the uncertainty in the VSSI produced by the 1991 Pinatubo eruption. Systematic uncertainties in the transfer functions for extra-tropical eruptions, based currently mostly on the climate model simulations of Gao et al. (2007) are difficult to quantify, but

likely larger than those of tropical eruptions.

Random errors in the VSSI estimates may be present because finite sampling: both the estimation of the ice sheet average flux from finite sample of ice cores, and because the ice sheets sample only a portion of the overall hemispheric deposition, which might vary from case to case because of the atmospheric transport and deposition of sulphate. Using standard error propagation rules, the random error in the total sulphur injection $\sigma_{M_S}$ due to random errors in the ice sheet composites and

the transfer functions is given by:

$$\sigma_{M_S} = \sqrt{\left(\frac{L^G f^G}{3}\right)^2 \left[\left(\frac{\sigma_{L^G}}{L^G}\right)^2 + \left(\frac{\sigma_{f^G}}{f^G}\right)^2\right] + \left(\frac{L^A f^A}{3}\right)^2 \left[\left(\frac{\sigma_{L^A}}{L^A}\right)^2 + \left(\frac{\sigma_{f^A}}{f^A}\right)^2\right]} \qquad (7)$$

The relative uncertainties in the composite ice sheet fluxes ($\sigma_{f^G}$, $\sigma_{f^A}$) are as described in Sec. 2.1.2. Estimates of the random error of the transfer functions ($\sigma_{L^G}$, $\sigma_{L^A}$) is presently impossible to estimate from observations. Model simulations of volcanic sulphur injection and its evolution suggest that the proportion of sulphur injected into the stratosphere and later

deposited to ice sheets can vary substantially due to variations in the meteorological state (Toohey et al., 2013). We take



model-based estimates of this variability, quantified at the $1\sigma$ level as 16 and 9 % for Greenland and Antarctica, respectively, as present best estimates of this representativeness error (Toohey et al., 2013).

## 2.4 Aerosol optical depth estimation

The Easy Volcanic Aerosol (EVA) version 1 forcing generator (Toohey et al., 2016b) is used here to translate sulphur injections into spatio-temporal resolved estimates of the optical properties of volcanic aerosols. We focus here on the variation of stratospheric aerosol optical depth (SAOD) at the mid-visible wavelength of 550 nm.

The EVA module takes stratospheric sulphur injection estimates as input, and outputs vertically and latitudinally varying aerosol optical properties designed for easy implementation in climate models. The spatiotemporal structure of the EVA output fields is based on a simple 3-box model of stratospheric transport, with timescales of mixing and transport based on fits to satellite observations of the 1991 Pinatubo eruption. Vertical and horizontal shape functions are assigned to each of the three boxes, again based on the observed extinction of Pinatubo aerosols. Internally, EVA calculates first the transport of sulphate mass between the three regions, then applies a scaling procedure to translate sulphate mass into mid-visible SAOD. This scaling is linear for most eruptions, and is based on retrievals of SAOD and total sulphur injection from the 1991 Pinatubo eruption. Following ICI, a nonlinear scaling is applied for very large eruptions: in EVA, the nonlinear scaling applies only to eruptions greater in magnitude than Tambora. EVA allows also for the consideration of a constant, non-zero stratospheric sulphur injection, representing the cross tropopause transport of naturally produced gases including carbonyl sulphide (Crutzen, 1976; Kremser et al., 2016) which gives rise to the "background" stratospheric sulphate aerosol layer (Junge et al., 1961). The smallest satellite-observed SAOD, which occurred around the year 2000, is used to estimate a constant background sulphur injection of 0.2 Tg yr$^{-1}$, which agrees well with other estimates (Sheng et al., 2015).

The SAOD results shown hereafter, produced by the EVA forcing generator using the eVolv2k VSSI database, are denoted as "EVA(2k)". SAOD results are shown in terms of either monthly, annual or centennial means—it should be noted that peak SAOD values can vary substantially depending on the temporal resolution of the record.

## 2.5 Comparison data sets

### 2.5.1 Ice sheet composite sulphate flux

The ICI reconstruction (Crowley and Unterman, 2013) provides ice core-based composite fluxes for the NH and SH over the period 800-2000 CE. SH fluxes are based on core records from Antarctica, while NH fluxes come from Greenland cores plus one core from Mt. Logan, Alaska. We rescaled the sulphate flux reported for Laki (1783) to undo the correction applied by the authors to account for tropospheric versus stratospheric injection by dividing by a factor of 0.15. The IVI2 database (Gao et al., 2008) does not directly provide ice core composite fluxes. However, by inverting the scaling procedure described by Gao et al., (2007), we have reproduced hemispheric composite fluxes based on the reported estimates of stratospheric



sulphate aerosol injection over the period 500-2000. We validated these results by comparing with the few sample fluxes reported by Gao et al. (2007).

### 2.5.2 Volcanic stratospheric sulphur injection

Global VSSI estimates are extracted from the IVI2 database (Gao et al., 2008) by first summing the reported hemispheric stratospheric sulphate aerosol injections from 501-2000. The sulphate aerosol masses of IVI2 are computed assuming 25% water content, so multiplication by a factor of 0.75 is required to convert sulphate aerosol mass into sulphate mass. Finally, conversion from sulphate mass to sulphur mass is computed based on the ratio of molecular weights.

The VolcanEESM database (Mills et al., 2016; Neely and Schmidt, 2016) contains estimates of total $SO_2$ emissions by volcanic eruptions from 1850-2015. For the pre-satellite era (1850 to 1979), the dataset combines the most recent volcanic sulphate deposition datasets from ice cores with volcanological and, where applicable, petrological estimates of the $SO_2$ mass emitted as well as historical records of large-magnitude volcanic eruptions. In the satellite era, volcanic emissions were primarily derived from remotely sensed observations. The database includes also the locations of each eruption, as well as estimates of the maximum and minimum plume height. To estimate the mass of sulphur injected into the stratosphere, we take the estimated plume heights and compare to the climatological tropopause height at the latitude of each eruption. If the maximum plume height is greater than the climatological tropopause, we assume that the $SO_2$ emitted is in fact injected to the stratosphere. $SO_2$ emissions from eruptions with maximum plume heights below the altitude of the local tropopause are thus ignored. Conversion from $SO_2$ to sulphur mass is performed by multiplication by the ratio of molecular weights.

### 2.5.3 Stratospheric aerosol optical depth

The ICI reconstruction (Crowley and Unterman, 2013) contains estimates of zonal mean SAOD at 550 nm for 4 equal area latitude bands over the period 800-2000. The reconstruction is based on a scaling of Greenland and Antarctic ice core composites to measured SAOD after the Mt. Pinatubo eruption of 1991. Here, we take the ICI SAOD estimates as they are provided, simply averaging the 4 equal-area bands into a global, annual mean SAOD.

For the 1850-2000 period, the CMIP6 stratospheric aerosol forcing reconstruction (Luo et al., in prep) has been constructed based on a combination of satellite and ground-based optical measurements, as well as aerosol-model results (Arfeuille et al., 2014) using VSSI estimates from IVI2. While the data set contains estimates of many physical and optical properties of the aerosols, we focus here on estimates of SAOD at 550 nm. Aerosol extinction from the CMIP6 forcing files are integrated above the climatological tropopause at each latitude, and a simple area-weighted average is used to compute the global mean.



## 3 Results

### 3.1 Ice sheet sulphate flux composites

Greenland and Antarctic composite sulphate fluxes from eVolv2k are compared to composites from the IVI2 (Gao et al., 2008) and ICI (Crowley and Unterman, 2013) reconstructions in Figure 1. For this comparison, we have focused on

unambiguous matches between the three sets of composites between ~1590 and 2000, also including the 1257/8 Samalas signal, listed in Table S2. Flux composites for eVolv2k and the IVI2 datasets are plotted against the ICI reconstruction in Figure 1.

The eVolv2k composite fluxes show rather close agreement with the values reported by ICI. Linear fits of the eVolv2k vs. ICI composite fluxes were computed included using the OLS bisector method (Isobe et al., 1990), resulting in slopes of 0.89

± 0.14 for Greenland and for 0.87 ± 0.12 Antarctica. On the other hand, the IVI2 flux values show a significant bias compared to ICI, with a slope of 1.33 ± 0.14 for GL and 1.30 ± 0.26 for Antarctica. Fits of the IVI2 fluxes against the eVolv2k fluxes (not shown) result in bias estimates of 1.49 ± 0.21 for Greenland and 1.50 ± 0.28 for Antarctica.

The apparent bias in the IVI2 fluxes compared to ICI and eVolv2k is primarily due to the reported fluxes for the largest events. When the linear fits are repeated after removing the largest events (1783, 1258 and 1815 for Greenland, 1258 and

1695 for Antarctica), the bias of the IVI2 fluxes compared to eVolv2k reduces to 0.90 ± 0.34 for Greenland and 1.1 ± 0.20 for Antarctica. For Greenland, IVI2 used a large number of supplemental ice cores in the estimates for Tambora and Laki (Clausen and Hammer, 1988; Mosley-Thompson et al., 1993), which increased the composites estimates for these events significantly compared to values originally reported using only the long-term ice core records (Gao et al., 2006). Over Antarctica, the large IVI2 composite flux for 1257 is likely a result of the very strong flux recorded by the SP01 ice core

from South Pole (Budner and Cole-Dai, 2003), which was not reproduced by another ice-core record from the same site (SP04, Ferris et al., 2011) and consequently was not included in the AVS-2k composite (Sigl et al., 2014).

### 3.2 Volcanic stratospheric sulphur injection

The eVolv2k global VSSI time series is shown in Figure 2, in comparison to the values from the IVI2 reconstruction. Over

the 1500-1900 time period, the two reconstructions are very similar in terms of the timing and magnitude of most events, including Tambora (1815) and Laki (1783). The eVolv2k VSSI estimates for Huynaputia (1600) and Parker (1640) are slightly larger than those of IVI2.

Within the 1000-1500 CE time period, the two reconstructions agree reasonably well in terms of the timing and magnitude of the great 1257 Samalas eruption, and the eruptions of 1276 and 1286. A major difference between the reconstructions is the

timing of the great mid-15th century eruption, which differs by 6 years. Before ~1250 CE, there is a notable shift in the timing of events, reaching about 6 years, and most events are of somewhat larger magnitude in the eVolv2k reconstruction. Between 500 and 1000 CE, there is very little correlation between the two reconstructions. The eVolv2k reconstruction



contains strong VSSI events, including events at 540, 574, 682, and 1108 CE which are missing or largely underestimated in the IVI2 reconstruction. The extension of VSSI estimates back to 500 BCE reveals two large events: a Samalas-magnitude injection in 426 BCE and an event of ~40 Tg sulphur—30% greater than Tambora—in 44 BC.

Global VSSI magnitudes for the major events common to the eVolv2k and IVI2 data sets over the 500-1900 CE period are

compared in more detail in Figure 3. Major events were defined here as those with VSSI values greater than 10 Tg [S], and lists were compiled of major events from both datasets, and matches between the two datasets were found based on coincidence in time (Table S3), allowing for a drift in time in the early portion of the overlap period consistent with recent updates to the ice core dating (Sigl et al., 2015). If no match was found for a strong event in one data set, a VSSI value of 0 was specified for the other data set.

VSSI estimates from eVolv2k and IVI2 for many of the largest events, including the 1257, 1783, and 1815 events, agree to ~10% (Fig. 3). This agreement reflects compensation between changes in ice core sulphate flux composites used in the two reconstructions (see Fig. 1) and the 33% increase in effective transfer function used in the construction of eVolv2k. If the 1452 event from IVI2 was matched to the 1458 event of eVolv2k, they would also agree to within 10%, yet this agreement is largely coincidental, since the IVI2 value is based on combining values from two likely disparate events in 1452 and 1458

(Cole-Dai et al., 2013; Plummer et al., 2012; Sigl et al., 2013). A handful of other smaller events agree between the two data sets to within 33%, including events in 536, 1182, 1276, 1286 and 1835. Five events (1230, 1171, 1600, 1640, and 1809, CE) have VSSI values 33-40% larger in eVolv2k, representing the impact of the increased transfer functions on similar ice sheet composite fluxes. Around 10 events in eVolv2k have VSSI values significantly more than 33% larger than in the IVI2 reconstruction. Some of these events appear to be missing (682, 1108) or significantly underestimated (540, 574, 626, 939)

in IVI2, likely due to a lack of synchronization of the underlying ice core records. Relative increases of 33-60% for other events (e.g., 1695 and 1831) reflect in part the identification of bipolar ice core signals, and therefore assignment of a tropical source for the eruption rather than an extratropical source assumed by IVI2.

Estimated random uncertainties in the VSSI values are displayed as vertical error bars in Fig. 3. Uncertainties for VSSI greater than 20 Tg [S] range from about 15-30 % (Fig. S6). Due to relatively uniform sulphate fluxes over the Greenland and

Antarctica ice core samples, VSSI estimates for the 1458 event and Tambora (1815) are among the most tightly constrained, with uncertainties of 15 and 16%, respectively. The VSSI uncertainty for Samalas (1257) is 18%, while the large events of 540 CE and Laki (1783) have larger uncertainties, with estimated values of 24 and 34%, respectively.

Centennial-scale variations in the eVolv2k and IVI2 VSSI reconstructions are shown in Figure 4. Centennial average VSSI values (Fig. 4a) are dominated by the largest events: maximum centennial averages occur in the 6th, 13th and 19th centuries,

which together contain 7 of the top 20 VSSI events of the eVolv2k data set (Table 2). Minimum centennial mean VSSI is found during a "Roman Quiet Period" in the 1st Century CE, with a mean value of about 0.1 Tg [S] yr$^{-1}$, a full order of magnitude less than that of the maximum century (1200-1300 CE) and less than a third of the long term average. The century with the second lowest level of volcanism is 1000-1100 CE, corresponding with the "Medieval Quiet Period" (Bradley et al., 2016). Compared to IVI2, eVolv2k VSSI averages are larger for all centuries, with large differences occurring near the



beginning of the period of overlap (e.g., the 6th, 7th, and 10th centuries), but also in the more recent centuries (e.g., 17th and 19th Centuries). The eVolv2k reconstruction also contains generally more events than that of IVI2 (Fig. 4b), with an average of 10.6 events per century, compared to 6.7 events per century in the IVI2 dataset. The centennial event frequency in eVolv2k is also slightly more uniform with time, with a coefficient of variation of 0.24, compared to 0.37 for IVI2. The

largest increase in the number of events identified in the eVolv2k database compared to IVI2 is in the years 500-1000 CE, when eVolv2k includes 12.0 events per century, compared to 5.5 events per century in IVI2.

### 3.3 Stratospheric aerosol optical depth

Time series of global mean SAOD from the EVA(eVolv2k) and ICI reconstructions are shown in Figure 5. (Zonal mean SAOD is shown for the full EVA(2k) reconstruction in Fig. S7). Similar to the VSSI comparisons, the timing and

magnitudes of major SAOD perturbations in the two reconstructions are similar from 1250 to 1900 CE, and significantly different before around 1200 CE.

A comparison of the magnitude of matched strong events in the eVolv2k and ICI SAOD reconstructions (Table S4) is shown in Fig 6. Maximum values of 3-yr cumulative SAOD are compared, to reduce differences due to the different temporal evolutions or assumed starting dates of events in the two reconstructions. Most of the largest SAOD events agree to within

33%, including the 1230, 1257, 1458, 1600, 1640, 1809 and 1815 events. Tambora is a notable case, with EVA(2k) cumulative SAOD approximately 25% smaller than that of the ICI reconstruction. Laki (1783) and other NH extratropical eruptions (e.g., 939, 1182) have much larger SAOD in the eVolv2k reconstruction, a result of not applying a correction for effusive, tropospheric eruptions as done in the ICI. Other apparent outliers can be understood to be a result of the inclusion of the then unsynchronized Plateau Remote and Taylor Dome ice cores in the ICI reconstruction. Contributing with 40-50%

weight to the mean Antarctic $SO_4$ flux composite before 1200 CE, the unsynchronized series from these two ice cores reduced in general the mean sulphate values for real volcanic events, while falsely generating apparent volcanic signals not observed by those ice cores that had been correctly synchronized (see Sigl et al., 2014, SOM for details).

Centennial mean SAOD estimates (Fig. 7) show larger mean values for EVA(2k) compared to ICI in all centuries. This difference is almost entirely due to the inclusion of a non-zero background SAOD in the EVA(2k) reconstruction: an

alternate version with no background sulphur injection, EVA(2k,nb), shows closer agreement with the centennial scale variation of ICI. The much stronger mean SAOD in the EVA(2k) reconstruction in the 18th century can be understood to result from the much larger estimate for Laki (1783). The stronger estimated mean SAOD for the 800-1200 time period is due to the identification of a number of events that are missing or have much smaller estimates in the ICI reconstruction.

The 1850-1900 period is included in the long-term EVA(2k) and ICI SAOD reconstructions, as well as the CMIP6 historical

(1850-2015) period forcing reconstruction, which is based on a mixture of observations and aerosol model results using VSSI estimates of the IVI2 reconstruction (Arfeuille et al., 2014). Over the overlapping 1850-1900 period, the EVA(2k) and CMIP6 reconstructions agree to within 20% in their estimation of the cumulative global mean SAOD for the 1883 Krakatau eruption (Fig. 8). In contrast, the ICI reconstruction's cumulative SAOD for Krakatau is 25% larger than that of the EVA(2k)



and CMIP6 estimates. Based on matching Greenland and Antarctic sulphate signals, eVolv2k and ICI identify a signal in 1862 as resulting from a tropical eruption, and the reconstructed SAOD in both reconstructions is roughly twice as large as that of CMIP6, where it is assumed to be extratropical. Figure 8 also shows the consistency in the background SAOD reconstructed in the eVolv2k and CMIP6 data sets, in contrast to the zero-level background assumed in the ICI reconstruction. Zonal mean SAOD from the ICI, EVA(2k) and CMIP6 reconstructions are shown in Fig. 8 over the same 1850-1900 period. The EVA(2k) reconstruction includes a smooth latitudinal structure, based on the observed evolution of aerosol after the 1991 Pinatubo eruption (Toohey et al., 2016b), avoiding the strong localized gradients in SAOD present in the 4-band structure of the ICI reconstruction. As in the ICI reconstruction, the EVA(2k) SAOD also reproduces hemispheric asymmetry in the SAOD based on the ratio of ice core fluxes from both hemisphere, avoiding potential biases related to purely simulated aerosol transport, as for example appear to be creating a strong NH bias in the CMIP6 SAOD representation of the Krakatau eruption.

### 3.4 A long-term context to 20[th] Century volcanic forcing

Long-term mean VSSI and SAOD estimates from eVolv2k and other reconstructions are listed in Tables 3 and 4 for different time periods, allowing a comparison with recent estimates for the 20[th] century.

The overall long-term (500 BCE-1900 CE) mean VSSI from the eVolv2k reconstruction is 0.49 Tg [S] yr$^{-1}$. This estimate is consistent with the estimate from (Pyle et al., 1996), based on satellite observations and compiled estimates of global eruption frequencies over the last ~200 years, although their estimate discounted the impact of the very largest eruptions. A mean VSSI rate of 0.49 Tg [S] yr$^{-1}$ is about 2.5 times larger than the best estimate of the yearly natural "background" input of sulphur to the stratosphere by cross-tropopause transport of aerosols and their precursors (Sheng et al., 2015).

Over the common period of overlap (500-1900), the eVolv2k VSSI injection mean is ~50% larger than that of the IVI2 reconstruction. As discussed earlier, this is due to the identification of more moderate eruptions, especially early in the time period, and an enhancement of the estimated magnitude of a number of moderate events.

The VolcEESM 20[th] century mean VSSI is approximately equal to the IVI2 long-term mean, but about 25% less than the long-term eVolv2k mean VSSI. This difference is similar to the difference between the 20[th] Century and long-term means in the IVI2 reconstruction. The VolcEESM 20[th] century reconstruction therefore appears to be more consistent with eVolv2k than with the IVI2 reconstruction.

The long-term (500 BCE-1900 CE) mean SAOD in EVA(2k) is 0.014. A version with no background sulphur injection produces a mean of 0.010, therefore, major volcanic sulphur injections contribute approximately a two-thirds of the long-term mean SAOD. The EVA(2k) version with no background injection shows very close agreement with the long-term mean of the ICI SAOD reconstruction: the larger mean SAOD in eVolv2k compared to ICI can thus be understood to be the result of the background injection. The CMIP6 20[th] century mean SAOD is about 14% less than the long-term EVA(2k) reconstruction. The fact that the difference between 20[th] century and long-term mean SAOD is smaller than that for VSSI is expected due to the inclusion of the constant background sulphur injection in the SAOD reconstruction.





## 4. Conclusions and discussion

This paper presents a new reconstruction of the climatic influence of major volcanic eruptions over the time span from 500 BCE to 1900 CE. The eVolv2k reconstructions of volcanic VSSI and SAOD presented here represent, first and foremost, the result of improved dating, resolution and synchronization of ice core sulphate records from Antarctica and Greenland (Sigl et al., 2014, 2015). Given the improvements in methodologies used to date and synchronize ice core records—including automated synchronizing and absolute dating through matching of signatures of cosmogenic isotopes in ice cores and tree rings in the 8[th] century (Sigl et al., 2015)—the eVovl2k reconstruction can be confidently assumed to be a more accurate estimate of volcanic forcing compared to prior reconstructions, particularly for time periods before ~1250 CE. This assertion is supported by strong correlation between the newly compiled volcanic ice core records and instances of sudden large-scale cooling from dendrochronological climate reconstructions (Sigl et al., 2015).

The eVolv2k reconstruction provides the input data for climate model simulations which aim to include external climate forcing agents as far back in time as 500 BCE. Volcanic stratospheric sulphur injection (VSSI) estimates can be directly ingested by suitable aerosol climate models. Alternatively, the EVA forcing generator (Toohey et al., 2016b), can be used to produce stratospheric aerosol optical properties, including the stratospheric aerosol optical depth (SAOD) based on the VSSI record. Aerosol optical properties can then be used as boundary conditions for model simulations. The eVolv2k reconstruction is the recommended volcanic forcing for transient simulations within the Paleo-modelling Intercomparison Project (PMIP, Jungclaus et al., 2016), and therefore represents an update to reconstructions most often used in prior paleo-simulations, including the IVI2 (Gao et al., 2008) and ICI (Crowley and Unterman, 2013) forcing data sets.

The eVolv2k VSSI estimates, and the related SAOD perturbations produced via the EVA forcing generator, show broad agreement with the IVI2 and ICI reconstructions over the 1250-1900 period in terms of the magnitudes of the largest volcanic events. For VSSI, agreement between eVolv2k and IVI2 is the product of compensating differences, including generally smaller ice core flux estimates and a larger effective transfer function used to scale ice core sulphate fluxes into VSSI estimates. For the SAOD reconstructions, agreement between the EVA(2K) and ICI reconstructions reflects relative consistency in the ice core composites (after ~1250) constructed in both efforts and a related methodology, wherein observations following the 1991 Pinatubo eruption are used to define the scaling from ice core sulphate to SAOD. Before 1250 CE, the eVolv2k VSSI and SAOD reconstructions include a number of events which are comparatively underestimated or completely missing in prior reconstructions. These updated estimates promise improvements in the attribution of forced and unforced climate variability before 1250 CE.

In general, estimates of long-term mean VSSI and SAOD in eVolv2k are larger than prior reconstructions. For VSSI, this reflects an increase in the number of identified events, and an increase of the estimated magnitude for a number of moderate to-strong events. The relative increase in long-term mean SAOD compared to prior work is primarily due to the inclusion of a non-zero minimum (or background) SAOD, which is consistent with the minimum in stratospheric SAOD observed by satellite sensors around the year 2000 CE. The long-term estimates of VSSI and SAOD evolution give context to the best



current estimates of 20<sup>th</sup> Century volcanic forcing. An independent estimate of 20<sup>th</sup> Century mean VSSI is about 25% smaller than the long term eVolv2k mean. Assuming stationarity of global eruption frequencies, it is therefore more likely than not that the 21<sup>st</sup> Century mean volcanic forcing will be greater than that of the 20<sup>th</sup> century.

For the first time, the eVolv2k reconstruction provides estimates of the uncertainty in volcanic VSSI estimates, based on estimated uncertainty in the ice core sulphate composites and the error inherent in using ice sheet average fluxes as a proxy for the full hemispheric sulphate deposition (and therefore the hemispheric atmospheric sulphate loading). These error estimates depend on the number of ice cores used in the composite and the degree of variation seen between the ice cores. For most of the largest volcanic events, the estimated uncertainty is around 20-30%, while for smaller events the estimated error reaches values of >50%. Systematic errors are also significant, representing the possibility that VSSI or SAOD estimates are biased in the long term average. The construction of transfer functions from the NWT of the 1950's and 1960's and from the single data point of the 1991 Pinatubo eruption carries significant uncertainties regarding the injection magnitudes and injection heights. Observational estimates of the VSSI of Pinatubo have uncertainties of ~20% (Guo et al., 2004), and recent modelling studies have argued for an VSSI from Pinatubo half that of the usual estimates (e.g., Dhomse et al., 2014). An important uncertainty in the reconstruction of volcanic forcing stems from the inability to differentiate ice core sulphate from vast, stratospheric sulphate clouds with that from relatively local effusive eruptions. Icelandic eruptions with large effusive emissions, like Laki or the Bardabunga eruption (Schmidt et al., 2015) could theoretically lead to large sulphate fluxes to the Greenland ice sheets with little-to-no significant stratospheric injection. VSSI and SAOD estimates then based on the raw Greenland sulphate records would produce large overestimates of the global effects of the eruptions. This issue led Crowley and Unterman (2013) to mute the impact of Greenland sulfate fluxes for signals that could be attributed to Icelandic eruptions in the ICI reconstruction. On the other hand, the vertical distribution of the sulphur emissions by Laki are still under debate (Lanciki et al., 2012; Schmidt et al., 2012), and while it seems likely that the Greenland sulphate signal for Laki does contain some component related to tropospheric emissions, the proportion of tropospheric to stratospheric injections is very unclear. For these reasons, we have not implemented a correction to the VSSI estimates of Laki or other known or suspected Icelandic eruptions. Uncertainties in the scaling procedures used within the EVA forcing generator certainly add another level of uncertainty to the SAOD estimates: for example, uncertainty in the measured SAOD after Pinatubo translates directly into systematic uncertainty in the SAOD estimated by EVA.

Future work should be able to further refine estimates of VSSI and SAOD presented here. First, a larger network of high quality ice core sulphate records from Greenland should reduce random errors in the composite flux due to limited sampling. Reducing uncertainty in the transfer functions used to link atmospheric sulphate content and ice-core sulphate fluxes would greatly improve estimates of volcanic forcing. Studies with atmospheric models show some promise (Gao et al., 2007; Toohey et al., 2013), but inter-model differences in stratospheric sulphate evolution highlight substantial uncertainties in the physical processes controlling aerosol growth and transport (Zanchettin et al., 2016). Emerging techniques to differentiate sulphur from tropospheric versus stratospheric origin (e.g., Lanciki et al., 2012) offer potential strategies for reducing uncertainties in future volcanic forcing reconstructions. Finally, continued extension of ice core volcanic records to the





present should soon provide important information, since anthropogenic sulphate deposition over Greenland—contaminated over much of the 20[th] century by anthropogenic sulphur emissions—has now almost reached pre-industrial levels, allowing the detection of moderate volcanic eruptions in Greenland.

## 5 Acknowledgements

Matthew Toohey acknowledges support by the Deutsche Forschungsgemeinschaft (DFG) in the framework of the priority programme "Antarctic Research with comparative investigations in Arctic ice areas" through grant TO 967/1-1. Computations were carried out at the German Climate Computing Centre (DKRZ). This work benefitted greatly as a result of the authors' participation in the Past Global Changes (PAGES) Volcanic Impacts on Climate and Society (VICS) working
group.

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





**Appendix A: Greenland ice core flux uncertainty analysis**

Simple statistical models can often be useful tools for estimating biases and errors in measured data sets (Dunn, 1989; Toohey and Strong, 2007). Given the apparent lack of bias between the three Greenland ice core sulphate records used in the eVolv2k Greenland composite (Fig. S4), we assume here a simple model, wherein the sulphate flux recorded by a single ice core ($x_i$) and its relationship with the ice sheet wide average flux ($\tau_i$) can be written

$$x_i = \tau_i + \delta_i \tau_i \tag{A1}$$

where $\delta_i$ is assumed to be a zero-mean, normally distributed random variable with error variance $\sigma_\delta^2$. We assume here that the measured values $x_i$ have constant relative errors, thus the total error ($\delta_i \tau_i$) is proportional to the ice sheet average $\tau_i$. Further measurements, e.g., from other sites on the ice sheet, can be similarly modelled:

$$y_i = \tau_i + \varepsilon_i \tau_i \tag{A2}$$

$$z_i = \tau_i + \gamma_i \tau_i \tag{A3}$$

Under this model, the variance of each measurement set can be written:

$$\sigma_x^2 = \sigma_\tau^2 + \mu^2 \sigma_\delta^2 + \sigma_\delta^2 \sigma_\tau^2 \tag{A4}$$

where $\mu$ is the expected value of $\tau_i$. The covariance between two measurements sets—assuming no correlation between the random errors—is given by:

$$\sigma_{xy} = \sigma_\tau^2 \tag{A5}$$

Solving Eq. A4 for $\sigma_\delta^2$, and replacing the population variances $\sigma_x^2$ and $\sigma_{xy}$ with the sample variances $s_{xx}$ and $s_{xy}$, produces an expression for the estimated error variance $\sigma_\delta^2$:

$$\hat{\sigma}_\delta^2 = \frac{s_{xx} - s_{xy}}{\mu^2 - s_{xy}} \tag{A6}$$

Equivalent expressions can be constructed for $\hat{\sigma}_\epsilon^2$ and $\hat{\sigma}_\gamma^2$. The covariance terms $s_{xy}$, $s_{xz}$, and $s_{yz}$ all act as estimates of the true ice sheet average variance $\sigma_\tau^2$, but with finite sample sizes, will give different values. A conservative estimate for each of the measurement error variances is produced by using the minimum values from the three covariances, which we have done here. Similarly, we use the minimum value from the means of $x$, $y$ and $z$ as the conservative estimate of $\mu$.

We used Eq. 6 (and the equivalent expressions for $\hat{\sigma}_\epsilon^2$ and $\hat{\sigma}_\gamma^2$) to estimate error variances for the Greenland NEEM, NGRIP and GISP2 data sets, based on volcanic events with sulphate flux values in all 3 cores. Since variances and covariances can be very sensitive to the largest values within their input fields, we computed the error variances iteratively, beginning with



the full dataset, and for each iteration, removing the event with the largest mean flux value over the three cores. The resulting error variance estimates fluctuate considerably after the removal of the first few largest values, then reach relatively stable values (Figure S5). We took the mean values for each error estimate over the n range of 3-11, over which the estimates are rather stable, but utilize almost the full data set. This analysis resulted in estimates for $1\sigma$ errors of 46% for NEEM, 45% for

5   NGRIP, and 33% for GISP2.





Table 1: Proposed matches of ice core sulphate signals to volcanic eruptions. Matches are based on those listed by Sigl et al. (2013), with appropriate time shift applied due to updated ice core timescales (Sigl et al., 2015), except where noted. Volcano names, eruption dates and eruption numbers are taken from the Volcanoes of the World database provided online by the Global Volcanism Program (2013), except where noted.

| Ice year | Eruption year | Eruption month | Eruption day | Volcano latitude | Volcano name | VEI | GVP Eruption number |
|---|---|---|---|---|---|---|---|
| 1887 | 1886 | 6 | 10 | -38.12 | Okataina (Tarawera) | 5 | 14506 |
| 1884 | 1883 | 8 | 27 | -6.10 | Krakatau | 6 | 15589 |
| 1875 | 1875 | 4 | 1 | 65.03 | Askja[1](Öskjuvatn Caldera) | 5 | 12911 |
| 1873 | 1873 | 1 | 8 | 64.40 | Grímsvøtn | 4 | 12818 |
| 1862 | 1861 | 12 | 28 | 0.32 | Makian | 4 | 16685 |
| 1856 | 1856 | 9 | 25 | 42.063 | Hokkaido-Komagatake[1] | 4 | 18567 |
| 1853 | 1853 | 4 | 22 | 42.50 | Toya (O-Usu) | 4 | 18598 |
| 1836 | 1835 | 1 | 20 | 12.98 | Cosiguina | 5 | 15718 |
| 1832 | 1831 | - | - | 19.52 | Babuyan Claro | 4 | 16880 |
| 1823 | 1822 | 10 | 8 | -7.25 | Galunggung | 5 | 15718 |
| 1815 | 1815 | 4 | 10 | -8.25 | Tambora[2] | 7 | 16231 |
| 1783 | 1783 | 6 | 15 | 64.40 | Grimsvotn (Laki)[3] | 4 | 12809 |
| 1766 | 1766 | 4 | 5 | 63.98 | Hekla (Bjallagigar) | 4 | 12745 |
| 1756 | 1755 | 10 | 17 | 63.63 | Katla | 5 | 12674 |
| 1739 | 1739 | 8 | 19 | 42.69 | Shikotsu (Tarumai) | 5 | 18612 |
| 1721 | 1721 | 5 | 11 | 63.63 | Katla[1] | 5 | 12673 |
| 1708 | 1707 | 12 | 16 | 35.36 | Fujisan | 5 | 17452 |
| 1673 | 1673 | 5 | 20 | 1.38 | Gamkonora | 5 | 16584 |
| 1667 | 1667 | 9 | 23 | 42.69 | Shikotsu (Tarumai) | 5 | 18610 |
| 1641 | 1640 | 12 | 26 | 6.11 | Parker | 5 | 16694 |
| 1601 | 1600 | 2 | 17 | -16.61 | Huaynaputina | 6 | 11795 |





| 1595 | 1595 | 3 | 9 | 4.89 | Ruiz, Nevado del | 4 | 11279 |
| 1585 | 1585 | 1 | 10 | 19.51 | Colima | 4 | 10414 |
| 1512 | 1510 | 7 | 25 | 63.98 | Hekla | 4 | 12739 |
| 1477 | 1477 | 2 | 1 | 64.63 | Bárdarbunga (Veidivötn) | 6 | 12865 |
| 1258 | 1257 | 7 (±3) | - | -8.42 | Rinjani (Samalas)[4] | 7 | 20843 |
| 946 | 946 | 11 (±2) | - | 41.98 | Changbaishan[5] | 7 | 19644 |
| 939 | 939 | 4 (±2) | - | 63.63 | Katla (Eldgjá)[6] | 4 | 19938 |
| 879 | - | - | - | 64.63 | Bardarbunga (Vatnaöldur) | 4 | 12854 |
| 853 | - | - | - | 61.38 | Churchill[7] | 6 | 20422 |
| 236 | - | 3 | 15 (±20) | -38.82 | Taupo[8] | 6 | 14553 |

1 Newly proposed match.

2 Date of most explosive phase of eruption, from Sigurdsson and Carey (1989).

3 Date of most explosive phase of eruption, from Thordarson and Self (2003).

5  4 Attribution and date estimate from Lavigne et al. (2013).

5 Attribution from Sun et al. (2014). Date based on inference from historical documents (Hayakawa and Koyama, 1998; Xu et al., 2013).

6 Date (including estimate of season) from "Timing and consequences of the Eldgjá eruption, Iceland" by Clive Oppenheimer, Andy Orchard, Markus Stoffel, Sébastien Guillet, Christophe Corona, Michael Sigl, Ulf Büntgen

10  7 Attribution from Jensen et al. (2014).

8 Eruption season derived from dendrochronological evidence (Hogg et al., 2012).



Table 2: The top 20 eruptions from the past 2500 years in terms of volcanic stratospheric sulphur injection (VSSI) in the eVolv2k reconstruction. Matched stratospheric sulphur injections from the IVI2 reconstruction (Gao et al., 2008), when available, are included for comparison.

| Eruption | eVolv2k | | IVI2 | |
|---|---|---|---|---|
| | Year (BCE/CE) | VSSI (Tg [S]) | Year | VSSI (Tg [S]) |
| Samalas | 1257 | 59.4 | 1258 | 64.5 |
| | -426 | 59.3 | - | - |
| | -44 | 38.6 | - | - |
| | 1458 | 33.0 | 1459/1452 | 5.5/34.4 |
| | 540 | 31.8 | 541 | 10.6 |
| Tambora | 1815 | 28.1 | 1815 | 27.0 |
| | 682 | 27.2 | No match | 0 |
| | 574 | 24.1 | 567 | 3.3 |
| | 1230 | 23.8 | 1227 | 16.9 |
| | 266 | 21.9 | - | - |
| Grimsvotn (Laki) | 1783 | 20.8 | 1783 | 23.2 |
| | 1809 | 19.3 | 1809 | 13.5 |
| | 1108 | 19.2 | No match | 0 |
| Huaynaputina | 1600 | 19.0 | 1600 | 14.1 |
| | 536 | 18.8 | 529 | 16.2 |
| Parker | 1640 | 18.7 | 1641 | 12.9 |
| | 1171 | 18.0 | 1167 | 13.0 |
| | -168 | 17.2 | - | - |
| Katla (Eldgjá) | 939 | 16.2 | 933 | 8.0 |
| | 433 | 15.9 | - | - |



Table 3: Long-term average annual VSSI estimates from different reconstructions

| Time period | eVolv2k VSSI (Tg [S] yr$^{-1}$) | IVI2 VSSI (Tg [S] yr$^{-1}$) | VolcEESM VSSI (Tg [S] yr$^{-1}$) |
|---|---|---|---|
| -500-1900 | 0.49 | - | - |
| 501-1900 | 0.54 | 0.35 | - |
| 1901-2000 | - | 0.25 | 0.37 |

Table 4: Long-term average SAOD from different reconstructions

| Time period | EVA(2k) SAOD | EVA(2k,nb) SAOD | ICI SAOD | CMIP6 SAOD |
|---|---|---|---|---|
| -500-1900 | 0.014 | 0.010 | - | - |
| 801-1900 | 0.015 | 0.011 | 0.011 | - |
| 1901-2000 | - | - | 0.009 | 0.012 |




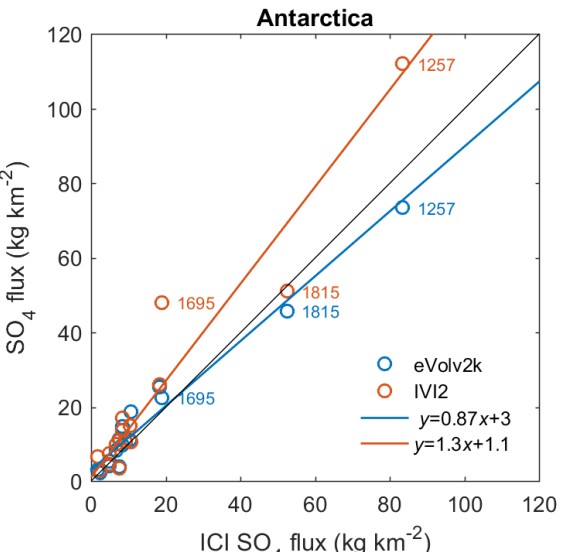
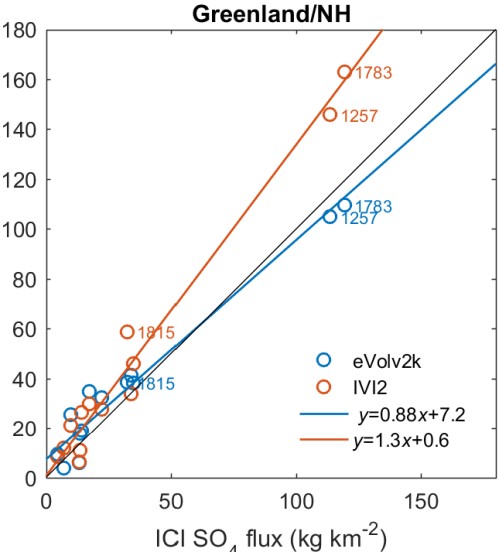

**Figure 1: Composite sulphate fluxes derived from ice cores for (left) Antarctica and (right) Greenland (or NH). Values from the eVolv2k (this work) and IVI2** (Gao et al., 2008) **reconstructions are plotted vs. composite values from the ICI reconstruction** (Crowley and Unterman, 2013)**, for event matches defined in Table S2. Linear fits to the scatter plots are included, with best fit slopes and intercepts as included in the legends.**





**Figure 2: Volcanic stratospheric sulphur injection (VSSI) from the IVI2 and eVolv2k reconstructions. Values exceeding the y-axis limits are denoted as text. Years are shown using the ISO 8601 standard, which includes a year zero.**





**Figure 3: Scatter plot of matched eVolv2k versus IVI2 VSSI estimates for events spanning 501-1900 CE with VSSI > 10 Tg [S]. Matches are defined in Table S3, and labels show the year of each event according to the eVolv2k reconstruction. Vertical bars indicate the 1σ uncertainty in the eVolv2k VSSI estimates. The 1:1 line is shown in black, with dark and light gray shading denoting the ±10 and 33 % range around 1:1.**



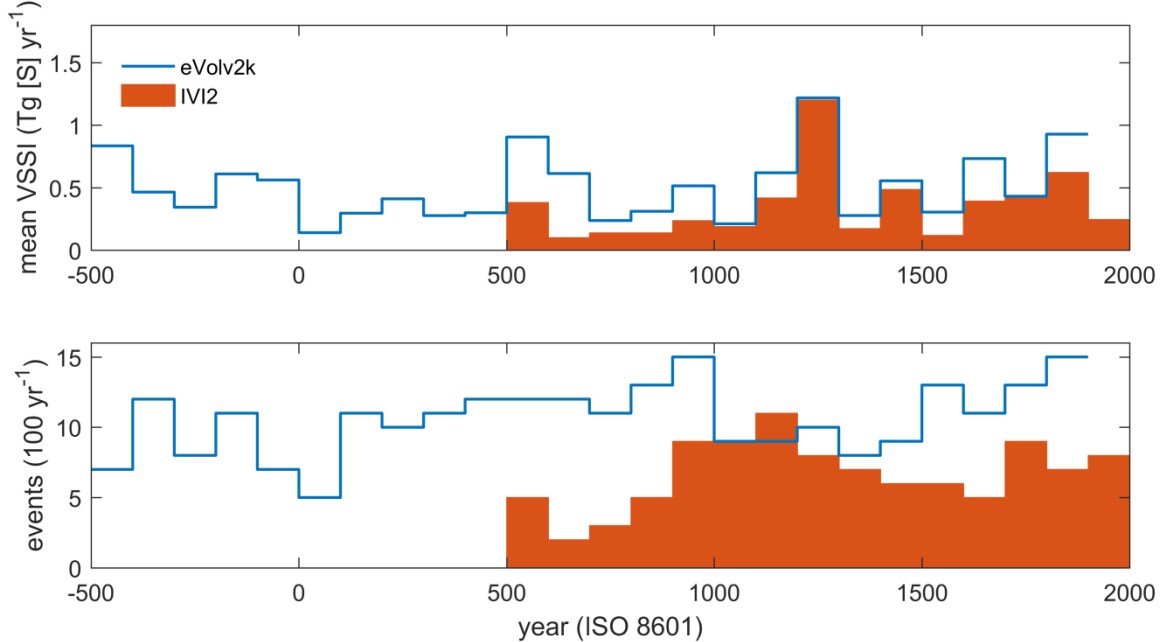

**Figure 4: (a) Centennial mean volcanic stratospheric sulphur injections (VSSI) from the IVI2 and eVolv2k reconstructions. (b) The number of volcanic events per century included in the eVolv2k and IVI2 reconstructions. Years are shown using the ISO 8601 standard, which includes a year zero.**



**Figure 5: Global mean, annual mean stratospheric aerosol optical depth (SAOD), from the EVA(2k) and ICI reconstructions. Years are shown using the ISO 8601 standard, which includes a year zero.**



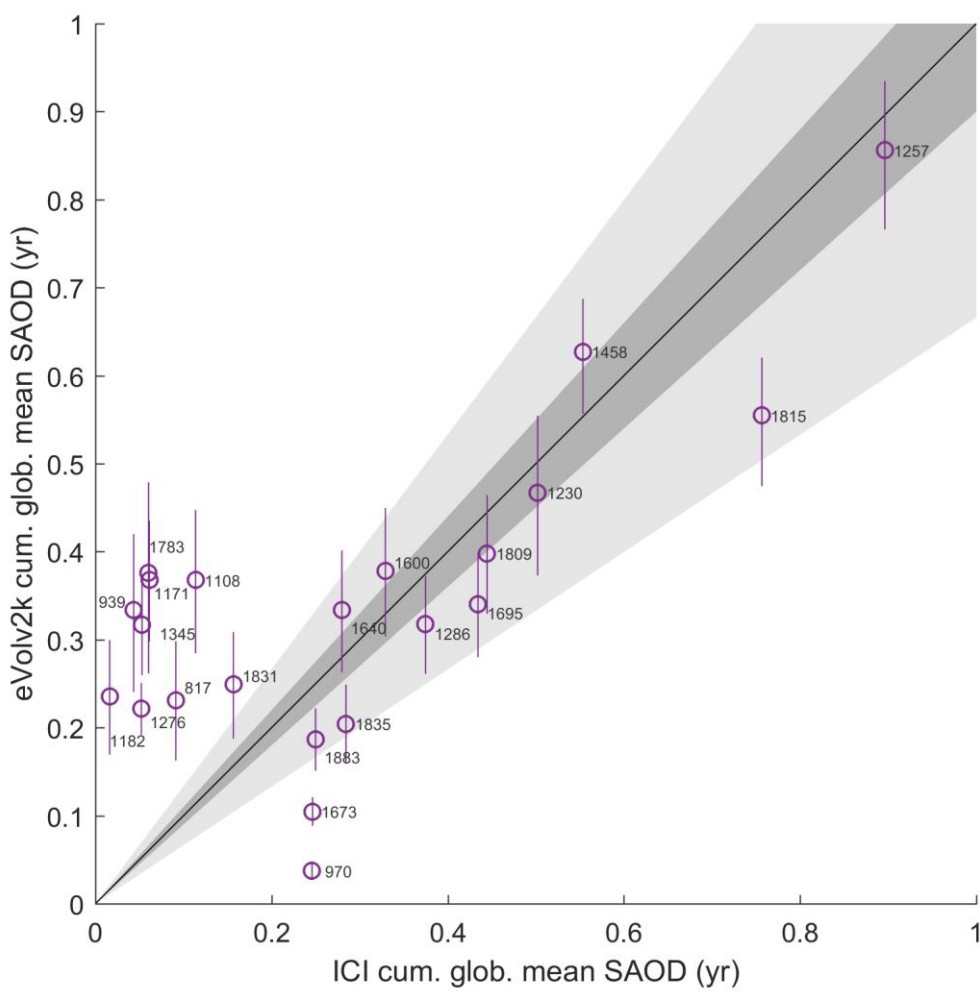

**Figure 6: Scatter plot of matched EVA(2k) versus ICI estimates of 3-year cumulative global mean SAOD, for events with values greater than 0.2. Matches are defined in Table S4. Labels show the eVolv2k date of each event. Vertical bars indicate the 1σ uncertainty in the EVA(2k) SAOD estimates. The 1:1 line is shown in black, with dark and light gray shading denoting the ±10 and 33 % range around 1:1.**





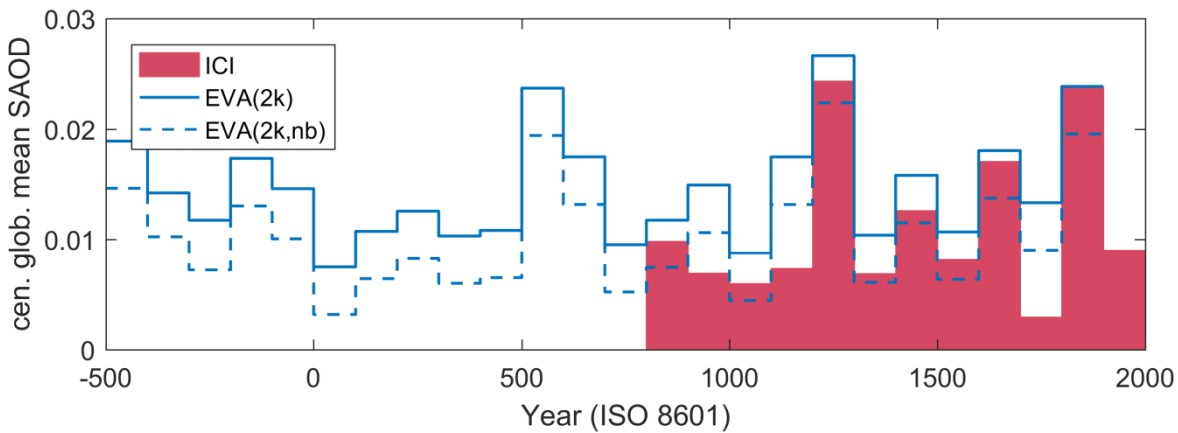

**Figure 7: Centennial global mean SAOD from the ICI and EVA(2k) reconstructions. A version of the EVA reconstruction which includes no background sulphur injection, EVA(2k,nb), is shown by the dashed line.**







**Figure 8: (top) Global mean SAOD from the ICI, EVA(2k) and CMIP6 reconstructions for the 1850-1900 time period of overlap. (bottom) Zonal mean SAOD from the three reconstructions as labelled for the same time period.**

