# Peer review of "Volcanic stratospheric sulphur injections and aerosol optical depth from 500 BCE to 1900 CE"

_Earth System Science Data, 2017_

## Referee Comment (RC1) · R.S. Bradley (Referee) · 7 Jul 2017

This is an important paper that will be widely read and cited by the paleoclimate community. It provides the (current) definitive record of explosive volcanism and estimates of the resulting stratospheric aerosol optical depth for the last 2500 years, based on a large set of ice core records from both Poles. The paper is clearly written, well-organised and unambiguous. It requires no revision, though I would have liked to see a bit more discussion of the events around 536CE (since they have received a lot of attention in the literature), & how the missing record from GISP2 might have affected the estimate of volcanism at this time. Similarly, there was nothing explicitly said about

the ∼1808/1809 eruption that preceded Tambora; previous studies have often ascribed this to a tropical eruption that might have contributed to the overall climatic impact of the subsequent 1815 event. In Table 2, this appears to be linked to Laki (Grimsvotn), yet I had the impression that there was a pretty big signal at that time in Antarctic records. . . Finally, I wonder why 45N & S is chosen for unknown extra-tropical eruptions – I would have thought that in the NH, the mean latitude of historic known explosive events is closer to 60N. Perhaps that could be checked and commented on. . .though it might not make a lot of difference to overall SAOD estimates.

I did not check all the citations, but noticed that Adolphi and Muscheler, 2016 was missing in the reference list, so perhaps there are others too....

––––––––––––––––––––––––––

---

## Referee Comment (RC2) · C. Gao (Referee) · 9 Jul 2017

This paper is an important contribution to the reconstruction of volcanic forcing for climate modeling studies. It builds on two previous ice-core-based reconstructions of global volcanism, by extending the temporal coverage to 500 BC and improving the event chronology. The paper is nicely written; the results are clearly presented and discussed. I recommend the paper to be published in this journal after addressing the following points:

1. A number of acronym have been introduced in the paper, which may cause some confusion without careful reading. To minimize the confusion, please consider use the

full name for certain terms (such as the EVA generator); for VSSI and SAOD, consider use eVolv2K-VSSI and eVolv2K-SAOD naming style consistently throughout the paper. 2. P6L20, please explain in more detail how does the analysis verify the representativeness of the few long term cores. In Figure 4, please use the number of 48 events (instead of the same marker) to plot the figure, so the readers could have a rough assessment of how individual event is represented. 3. P7L2-4, "Before 1 CE, ......, a constant uncertainty value of 26% is assumed, based on regression analysis between AVS-2k...and single ice-core records". Please specify on which period was the regression analysis done. 4. P7L8-11, please provide a brief description of the method of standard error propagation rules, and explain how the two-core and three-core composite uncertainties were obtained. Since the assessment of the signal core uncertainties were provide in Appendix A. The authors may consider add the assessment of the two-core and three-core composite uncertainties right afterward. 5. P8 section 2.3, the discussion of previous work leading towards the modification done by this study could be shortened or moved to the supplementary information. 6. P10L25-29, could the authors please provide a list of eruptions that belong to "such cases", and explain briefly how "such cases" were identified. 7. P17L24-26, I do not quite follow the logic of the discussion in this paragraph. When the authors came to the conclusion that "The VolcEESM 20th 25 century reconstruction therefore appears to be more consistent with eVolv2k than with the IVI2 reconstruction.", did they assume that the 25% decrease of 20th century VSSI from the 500-1900 CE mean VSSI in IVI2 also holds true in VolcEESM and eVolv2k? How was this assumption justified, if the magnitude of IVI2 itself were not in good agreement with the other two?

A few comments on the technical details: 1. P4L24 Introduce the name of these two chronologies (i.e., NS1-2011 and WD2014) here, so the readers will know that the NS1-2011 and WD2014 discussed later are not new chronologies. 2. P9L10, please change "(Gao et al., 2007) used" to "Gao et al. (2007) used". 3. P12L20, "The SAOD results shown hereafter, produced by the EVA forcing generator using the eVolv2k VSSI database, are denoted as 'EVA(2k)'". "EVA(2k) " does not seem to appear in the later

sections; instead, "EVA(eVolv2k)" was referred to in various parts of the paper. Could the authors please check whether the two terms refer to the same data, and fix it?

---

## Author Comment (AC1) · 11 Sep 2017

Response to Reviewers

We sincerely thank Professors Bradley and Gao for their comments on the initial manuscript, which have helped to improve and clarify the presentation of the data set. Below, reviewer comments are listed in black, and our replies and revisions in blue. Our page and line number references refer to the "tracked changes" version of the revised manuscript.

Reviewer 1: Raymond Bradley

This is an important paper that will be widely read and cited by the paleoclimate community. It provides the (current) definitive record of explosive volcanism and estimates of the resulting stratospheric aerosol optical depth for the last 2500 years, based on a large set of ice core records from both Poles. The paper is clearly written, well organised and unambiguous. It requires no revision, though I would have liked to see a bit more discussion of the events around 536CE (since they have received a lot of attention in the literature), & how the missing record from GISP2 might have affected the estimate of volcanism at this time.

We have a great interest in the events around 536 CE. The major reason for confusion regarding the volcanic origin of the NH temperature anomalies at the time is primarily a result of errors in the dating of ice cores, recently corrected by Sigl et al. (2015). We have added the following statement regarding the 6th century (P15, L12-15):

*Of particular note, the eVolv2k reconstruction includes a sequence of very large eruptions in the 6th Century, including a NH extratropical eruption in 536 CE, and tropical eruptions in 540 and 574 CE, consistent with timings inferred in earlier studies (Baillie, 2008; Baillie and McAneney, 2015; Toohey et al., 2016a) and confirmed by (Sigl et al., 2015), with VSSI magnitudes 25-30% larger than estimated by (Toohey et al., 2016a).*

Similarly, there was nothing explicitly said about the 1808/1809 eruption that preceded Tambora; previous studies have often ascribed this to a tropical eruption that might have contributed to the overall climatic impact of the subsequent 1815 event.

We have added another mention of 1809 as one of the cases where the eVolv2k VSSI reconstruction is larger than that of the IVI2 reconstruction (P15, L4-5). However, the SAOD estimate from eVolv2k is rather similar to that of the ICI AOD reconstruction, so we have chosen not to include a large discussion in this work of any potential change in our understanding of the combined impacts of 1809 and Tambora.

In Table 2, this appears to be linked to Laki (Grimsvotn), yet I had the impression that there was a pretty big signal at that time in Antarctic records…

Table 2 lists Laki (Grimsvotn) as the source of the 1783 ice core signal. 1809 remains unidentified. In order to avoid misreading in Table 2, we have added "Unidentified" for appropriate cases rather than blank space.

Finally, I wonder why 45N & S is chosen for unknown extra-tropical eruptions – I would have thought that in the NH, the mean latitude of historic known explosive events is closer to 60N. Perhaps that could be checked and commented on… though it might not make a lot of difference to overall SAOD estimates.

In the EVA forcing generator, the exact latitude makes no impact on the estimated forcing beyond a separation between the 3 defined regions: tropical (-25<φ<25), extratropical NH (φ>25) and extratropical SH (φ <-25). That being said, we agree that some degree of accuracy to the assumed latitudes for unknown eruptions would be a good thing. From the GVP Holocene database, we selected extratropical (φ >25) events with VEI>=4, and find that the average latitude for NH extratropical events is ~48°N, and the average for SH events is 42°S. We now include a statement of the GVP mean latitudes in the main text, so that users may replace the default 45°N and 45°S values if they wish, however, we will keep the default values as they are, as there is some logic to maintaining symmetric injection latitudes in the two hemispheres as the default assumption, especially given the uncertainties in the sampling of the GVP database.

Revision (P8, L2-5):

*Representative latitudes for unidentified eruptions can be based on the latitudinal distribution of identified eruptions: based on the Holocene eruption database (Global Volcanism Program, 2013), we find average latitudes of extratropical (|φ|>25), VEI ≥ 4 eruptions of 48°N and 42°S, and an average tropical eruption latitude of 2°N. For simplicity and symmetry, we adopt a convention of assigning unknown tropical eruptions a latitude of 0°N, and extratropical eruptions a latitude of 45°N or S. Consistent with Crowley and Unterman (2013) unknown eruptions are assigned an eruption date of January 1.*

I did not check all the citations, but noticed that Adolphi and Muscheler, 2016 was missing in the reference list, so perhaps there are others too....

OK, thank you, we have double checked the references and found a handful of other missing references which have now been added to the list.

Reviewer 2: Chaochao Gao

This paper is an important contribution to the reconstruction of volcanic forcing for climate modeling studies. It builds on two previous ice-core-based reconstructions of global volcanism, by extending the temporal coverage to 500 BC and improving the event chronology. The paper is nicely written; the results are clearly presented and discussed. I recommend the paper to be published in this journal after addressing the following points:

1. A number of acronym have been introduced in the paper, which may cause some confusion without careful reading. To minimize the confusion, please consider use the full name for certain terms (such as the EVA generator); for VSSI and SAOD, consider use eVolv2K-VSSI and eVolv2K-SAOD naming style consistently throughout the paper.

We appreciate that the naming system used here is a little unorthodox. Out motivation for naming the SAOD reconstruction "EVA(eVolv2k)" is to clearly separate the two main steps in the reconstruction methodology, and reinforce the idea that EVA can be used to produce SAOD from any sulfur injection reconstruction, so one could compare e.g., EVA(eVolv2k) with EVA(GAO08), the latter using VSSI estimates from Gao et al., 2008, or EVA(VolcanEESM) to use another example. While in the present paper, we agree that terms "eVolv2K-VSSI" and "eVolv2K-SAOD" would be clear, we hope to establish the EVA(X) naming convention, which will be useful in other contexts.

We have added the following sentence to the end of Sec 2.4 (P12, L28-29):

*This naming convention is used to emphasize the two-step nature of the SAOD reconstruction, and encourage clarity in future cases when, e.g., EVA is used with other input data sets.*

2. P6L20, please explain in more detail how does the analysis verify the representativeness of the few long term cores. In Figure 4, please use the number of 48 events (instead of the same marker) to plot the figure, so the readers could have a rough assessment of how individual event is represented.

The reliability of single ice-cores to capture a representative sample of Earth's past atmospheric sulfate content was recently questioned on the basis of replicate ice-core measurements (Gautier et al., 2016). Their results we believe resulted from a very unlucky sampling site selection on the dry (<3 cm ice equivalent of annual snowfall) and wind-exposed plateau. Their measurements indicated that the Tambora signal was absent in some of their replicate ice-cores. Using a much wider ice-core network and the depositional records of many more eruptions we find that all major volcanic signals are recorded in virtually all the ice core records (providing data is available for the respective sections). We further found that stacking two records only (at least when sampled at high-resolution sites not subject to strong wind erosion) resulted in average sulphate deposition values that correlate strongly with the comprehensive record of >12 sites, thus much less records sampled from good sites are required to retrieve robust estimates of large-scale deposition. We added this information to the manuscript. We do not believe that identifying the individual events will bring in important information for the broad readership of this journal or improve the readability of the Figure.

We have revised a sentence to the following (P6, LL31):

*The high level of correlation - especially over Antarctica, where a large number of individual records is contributing to the composite - indicates that the lack of replication of the Tambora signal observed at Dome C (Gautier et al., 2016) is most likely a phenomenon specific to the wind-exposed Antarctic plateau, and that in general, individual ice-core records are able to capture a large portion of the large-scale sulphate deposition. In other words, the strong correlations between ice sheet composites and the single site records support the idea that valuable information on large-scale sulphate flux and therefore*

*stratospheric aerosol burdens can be extracted even from a small number of ice cores from well-placed sites.*

3. P7L2-4, "Before 1 CE,…, a constant uncertainty value of 26% is assumed, based on regression analysis between AVS-2k… and single ice-core records". Please specify on which period was the regression analysis done.

The regression analysis was performed between the AVS-2k record and the composite of B40/WDC over the period 1-2000 CE, this information has been added (P7, L7-8).

4. P7L8-11, please provide a brief description of the method of standard error propagation rules, and explain how the two-core and three-core composite uncertainties were obtained. Since the assessment of the signal core uncertainties were provide in Appendix A. The authors may consider add the assessment of the two-core and three-core composite uncertainties right afterward.

We have added a short explanation and formula to the end of Appendix A.

5. P8 section 2.3, the discussion of previous work leading towards the modification done by this study could be shortened or moved to the supplementary information.

We appreciate that this section is rather long and detailed, but we feel that in order to justify the rather large (~33%) increase in effective transfer function, it is best to give a thorough treatment of the derivation, and to clearly refer to the prior work that has pioneered this important method.

6. P10 L25-29, could the authors please provide a list of eruptions that belong to "such cases", and explain briefly how "such cases" were identified.

The only significant case (VSSI>3 Tg S) where this situation applies, is that of the Taupo eruption of 236 CE, with an Antarctic mean flux of 14 kg km$^{-2}$ and a Greenland flux of 9 kg km$^{-2}$. We have added this information to the text. Upon searching for such cases, we identified a few other minor eruptions attributed to the SH with corresponding signals in Greenland—these attributions may require revision, and will be re-evaluated in future versions of eVolv2k.

Revision (P11, L2-3): *In the current eVolv2k version, the only significant eruption for which this rule applies is the 236 CE Taupo event, for which sulphate signals in both Antarctica and Greenland are attributed to the SH eruption.*

7. P17L24-26, I do not quite follow the logic of the discussion in this paragraph. When the authors came to the conclusion that "The VolcEESM 20th 25 century reconstruction therefore appears to be more consistent with eVolv2k than with the IVI2 reconstruction.", did they assume that the 25% decrease of 20th century VSSI from the 500-1900 CE mean VSSI in IVI2 also holds true in VolcEESM and eVolv2k? How was this assumption justified, if the magnitude of IVI2 itself were not in good agreement with the other two?

We agree that this paragraph is hard to follow, and since it is not central to the conclusions of the study, we have decided to simply delete the paragraph.

A few comments on the technical details:

1. P4L24 Introduce the name of these two chronologies (i.e., NS1-2011 and WD2014) here, so the readers will know that the NS1-2011 and WD2014 discussed later are not new chronologies.

Thank you, we have done this as suggested.

2. P9L10, please change "(Gao et al., 2007) used" to "Gao et al. (2007) used".

Done, thank you for catching this typo.

3. P12L20, "The SAOD results shown hereafter, produced by the EVA forcing generator using the eVolv2k VSSI database, are denoted as 'EVA(2k)"'. "EVA(2k) " does not seem to appear in the later sections; instead, "EVA(eVolv2k)" was referred to in various parts of the paper. Could the authors please check whether the two terms refer to the same data, and fix it?

Yes, sorry, we modified our naming convention during the writing process and failed to update all occurrences. We have fixed this in the updated version.